# Microbial plankton occurrence database in the North American Arctic region: synthesis of recent diversity of potentially toxic and/or harmful algae

Nicolas Schiffrine[1], Fatma Dhifallah[1], Kaven Dionne[1, a], Michel Poulin[2], Sylvie Lessard[1], André Rochon[1], Michel Gosselin[1]

[1]Institut des sciences de la mer (ISMER), Université du Québec à Rimouski, Rimouski, QC G5L 3A1, Canada
[2]Recherche et collections, Musée canadien de la nature, C.P. 3443, Succ. D, Ottawa, ON K1P 6P4, Canada
[a]present address: Réseau Québec Maritime, Université du Québec à Rimouski, Rimouski, QC G5L 3A1, Canada

*Correspondence to*: Nicolas Schiffrine (nicolas_schiffrine@uqar.ca)

**Abstract**. The Arctic Ocean is currently undergoing significant transformations due to climate change, leading to profound changes in its microbial plankton communities, including photoautotrophic prokaryotes and eukaryotes (i.e., phytoplankton), as well as heterotrophic, phagotrophic, and mixotrophic protistan species. Among these unicellular organisms, potentially toxic and/or harmful algal species (hereafter referred to as "HA") are of particular concern, as they pose a threat to human and ecosystem health if they potentially spread into Arctic waters. Despite their importance, the spatial and temporal distribution of these communities in the North American Arctic is poorly understood. To address this gap, we compiled and synthesized a large dataset from various sources, partitioned into nine regions based on the Large Marine Ecosystem classification. Our dataset contains 385 348 georeferenced data points and 18 268 unique sampling events (Schiffrine et al., 2024), encompassing 1442 unique taxa, with Heterokontophyta (notably diatoms) and Dinoflagellata being the most dominant phyla. Our results indicate distinct spatial patterns of diversity, with the highest diversity observed in Atlantic-influenced regions of the North American Arctic. An analysis of the maximum latitude of HA species over time shows a gradual increase, with a notable rise towards the 1990s. However, this trend is likely influenced by increased research in higher latitudes, meaning no substantial spread of HA species into the North American part of the Arctic. Our study underscores the importance of extensive and long-term sampling efforts to understand the Arctic's biodiversity, particularly in documenting the presence and distribution of HA species. While the occurrence of HA in the Arctic is recognized, our findings highlight the need for further detailed investigations to fully grasp their ecological impacts and variability in the region. Overall, our results provide new insights into the spatial patterns and biodiversity of the microbial plankton communities in the North American Arctic and have implications for understanding the ecological functioning and response of this region to ongoing climate change.

## 1 Introduction

The Arctic Ocean has become a focal point for climate change research due to its vulnerability to rapid and significant alterations in the environment (Meredith et al., 2019). As a result, the Arctic has been the focus of a growing number of scientific investigations aimed at understanding how these transformations affect the region's ecosystems, people, and global climate. In 2007, the Intergovernmental Panel on Climate Change recognized the Arctic as a region among the most vulnerable to climate change, highlighting the urgent need for further research in this area. Since then, a large body of scientific literature has emerged that explores the effects and implications of climate change on the Arctic marine ecosystem.

The Arctic Ocean is undergoing rapid changes, with surface waters warming nearly four times faster than the global average (Rantanen et al., 2022), leading to significant reductions in sea ice extent and thickness (Hanna et al., 2021; Kacimi and Kwok, 2022). This has resulted in increased melt pond formation (Rösel et al., 2012) and changes in ice dynamics, affecting energy absorption and water column stratification (Carmack et al., 2016), thereby reducing nutrient supply (Yamamoto-Kawai et al., 2009). Additionally, the Arctic Ocean is experiencing some of the fastest rates of ocean acidification (AMAP, 2018), adding to the numerous challenges already faced by this rapidly changing region. These dramatic environmental changes are reshaping the microbial plankton communities, including photoautotrophic prokaryotes and eukaryotes (i.e., phytoplankton), as well as heterotrophic, phagotrophic, and mixotrophic protistan species, with significant implications for ecosystem structure and function (e.g., Ardyna and Arrigo, 2020). The reduction in sea-ice cover has led in earlier phytoplankton blooms in some regions (Kahru et al., 2011), and increased open water periods, potentially leading to a second bloom in autumn (Ardyna et al., 2014). In the Barents Sea, shifts in the current surface velocities have driven poleward intrusions of *Gephyrocapsa huxleyi* (Lohmann) Reinhardt (previously called *Emiliania huxleyi* (Lohmann) Hay & Mohler; Bendif et al., 2019, 2023), a temperate marine calcifying phytoplankton species (Neukermans et al., 2018; Oziel et al., 2020). Conversely, in the less productive waters of the Canada Basin, increased freshwater inflow has led to a transition from nanophytoplankton/diatom communities to picophytoplankton due to altered nutrient availability in the surface layer (Li et al., 2009). The observed changes in physico-chemical conditions in the Arctic may also increase the potential risk of proliferation of potentially toxic and/or harmful algal species (hereafter abbreviated as HA). Numerous HA species have already been detected in several Arctic regions (Bates et al., 2020; McKenzie et al., 2020). Notably, various toxin-producing diatoms of the genus *Pseudo-nitzschia* H.Peragallo have been documented in Iceland, Western Greenland, Baffin Bay, Barrow Strait, Beaufort Sea, Bering Strait, and subarctic regions around Norway (Bates et al., 2020; Pućko et al., 2019). Similarly, toxic dinoflagellate species belonging to the genera *Alexandrium* Halim and *Dinophysis* Ehrenberg have been detected (Bates et al., 2020; Bruhn et al., 2021; Dhifallah et al., 2021; Okolodkov and Dodge, 1996; Pućko et al., 2019). Olsen et al. (2019) recently documented a red tide of the harmful phototrophic ciliate *Mesodinium rubrum* (Lohmann) Leegard at the interface between ice and water in newly formed pack ice north of Svalbard during early spring. Their findings suggest that ephemeral blooms of this species are increasingly probable under the context of thinning Arctic sea ice. Nöthig et al. (2015) also described a dominance shift towards the harmful

prymnesiophyte *Phaeocystis pouchetii* (Hariot) Lagerheim, driven by a warm-water anomaly in the Atlantic waters of the West
Spitsbergen Current in Fram Strait. Moreover, the increase in maritime traffic due to growing economic and tourism
development in the Arctic may elevate the risk of introducing non-native species, including HA species (Chan et al., 2019;
Dhifallah et al., 2021). These shifts could have significant implications for the future of Arctic marine ecosystems, impacting
the transfer of energy and organic matter through the pelagic food web.

The paucity of data on the diversity and richness of Arctic microbial plankton communities, particularly phytoplankton and
other protist species, hinders our ability to fully understand their spatial and temporal variability. Additionally, the complex
biogeography of the polar region exacerbates this issue. One way to address these challenges and track potential changes in
community structure, dynamics, and phenology is through the use of long-term datasets. The emergence of digital archives of
biological data, such as the Global Biodiversity Information Facility (GBIF; https://www.gbif.org/) and the Ocean
Biogeographic Information System (OBIS; https://www.obis.org/), has enabled the identification of significant patterns in the
global distribution of microbial plankton diversity as well as the occurrence, toxicity, and associated risks posed by HA species
(Hallegraeff et al., 2021; Righetti et al., 2019). Despite numerous studies that have utilized long-term datasets to monitor
changes in Arctic microbial plankton diversity and dominance, most of these studies have been conducted in specific regions
of the Arctic (Blais et al., 2017; Freyria et al., 2021). Furthermore, previous reports on the diversity of Arctic microbial
plankton communities have not included essential information, such as geographic coordinates and dates, limiting the ability
to assess potential changes in diversity and dominance (e.g., Poulin et al., 2011). To date, there has been no effort to
comprehensively combine data from various sources, such as OBIS, GBIF, and both published and unpublished datasets, into
a unified database specifically for the North American Arctic. This study aims to fill this gap by creating an extensive database
on a pan-American Arctic scale. This database will facilitate the investigation of global trends in the biogeography, diversity,
and composition of microbial plankton taxa across the North American region of the Arctic Ocean, thereby addressing the
limitations of existing quantitative data.
**2 Data and methods**
**2.1 Data acquisition**
Our database consists of microbial plankton occurrences (i.e., presences and abundances greater than zero), including
photoautotrophic prokaryotes and eukaryotes (i.e., phytoplankton), as well as heterotrophic, phagotrophic, and mixotrophic
protistan species. These data were compiled from web-based search engines and queries in online databases, such as OBIS
(https://obis.org), GBIF (https://www.gbif.org) and PANGAEA (https://www.pangaea.de/). Occurrence data from OBIS (last
accessed: November 20, 2020; n = 575 200) and GBIF (last accessed: November 16, 2020; n = 197 439) were first downloaded
using the keywords "Chromista" and "Plantae"; from 45° N to 90° N and from 40° W to 180° W, without temporal restriction.
Occurrence data from PANGAEA (last accessed: November 2020; n = 1994) were collected using the keywords: "Chromista",
"Phytoplankton", "Taxonomy", "Harmful algal bloom", "Arctic Ocean", "Polar" and several combinations of these keywords.
We supplemented the data with records from ArcticNet campaigns (n = 43 982) and individual studies (n = 90 479). We
included the "sourceArchive" column (
) to specify the origin of each record (i.e., GBIF, OBIS, ArcticNet, or individual studies). We standardized the column names
to ensure compatibility between different datasets and to adhere to the Darwin Core standard (https://dwc.tdwg.org/), resulting
in a comprehensive dataset of 909 094 data points (Schiffrine et al., 2024).

**2.2 Biogeographic classification**

Our global database was divided into hexagonal bins using the R package *dggridR* (version 3.1.0; https://github.com/r-
barnes/dggridR; Barnes and Sahr, 2020), with a resolution of 2591.40183 km$^2$. The chosen grid resolution strikes a balance
between providing sufficient spatial resolution to capture ecological patterns and minimizing computational requirements.
Each grid cell was then assigned a corresponding Large Marine Ecosystem (LME) region using the spatial polygons obtained
from the "*mr_shp*" function of the R package *mregions* (version 0.1.9; Chamberlain and Schepers, 2021). Arctic LMEs are
defined by ecological criteria, including bathymetry, hydrography, productivity, and tropically linked populations, and are
integral to ecosystem-based management approaches with a 5-module framework focused on productivity, fish and fisheries,
pollution and ecosystem health, socioeconomics, and governance (PAME, 2013). Conserving only grid labelled as "arctic"
according to the LME classification (PAME, 2013), this new dataset contains 4458 grid cells partitioned into nine different
regions and 550 033 data points (Schiffrine et al., 2024).

**2.3 Data quality control**

Each record underwent a verification process to ensure the accuracy of taxonomic identification. First, we used the AlgaeBase
database, and the API key provided by the AlgaeBase team to validate each record as an accepted name
(http://www.algaebase.org/; AlgaeBase. World-wide electronic publication, 2023). If a record was not validated through this
process, we performed a secondary verification using the "*wm_records*" function from the R package *worrms* (version 0.4.3;
Chamberlain and Vanhoorne, 2023), using the World Register of Marine Species database (WoRMS;
http://www.marinespecies.org; Ahyong et al., 2023). If the taxonomic identification could not be found in either of these
databases, we assigned the record to the next higher taxonomic classification level (n = 39). These modifications were specified
in the "ReduceName" column (Table 1). In order to maintain data quality and avoid loss of information, we manually adjusted
a total of 249 taxonomic names, with the modified names indicated in the "ModifiedName" column (Table 1). Taxonomic
records that included qualifiers such as "aff." (n = 40) and "cf." (n = 95) were categorized at the species level in our dataset to
simplify taxonomic classification. While this simplification enhances dataset accessibility, it is crucial to acknowledge the
potential introduction of errors due to a certain degree of uncertainty associated with species identification. To maintain
transparency, qualifiers originally denoted by "cf." and "aff." were thoughtfully preserved in the "OpenNomenclature" column
(Table 1). This approach strikes a balance between simplification and taxonomic rigor, enabling users to recognize the initial
uncertainty in identification and facilitating further investigation or refinement of taxonomic assignments as necessary.
Taxonomic records with qualifiers such as "sp." (n = 193) or "spp." (n = 324), as well as those indicating a "group" (e.g.,
*Pseudo-nitzschia seriata* group; n = 27), "complex" (e.g., *Gymnodinium*/*Gyrodinium* complex; n = 3), or containing multiple
species names (e.g., *Pseudo-nitzschia delicatissima*/*Pseudo-nitzschia pseudodelicatissima*; n = 12), were categorized at the
genus level in the dataset. These qualifiers are denoted in the "OpenNomenclature" column (Table 1). Less than 1% of the
records in our dataset could not be identified in either the AlgaeBase or WoRMS databases. The original taxonomic names
were retained in the "verbatimScientificName" (Table 1), allowing for traceability to the harmonized names.

**Table 1: Detailed description of the columns. For details see: https://dwc.tdwg.org/**

| Variable name | Definition |
|---|---|
| acceptedNameUsage | Currently accepted name according to AlgaeBase and/or WoRMS |
| eventDate | Date and time of the event |
| year | Integer representing the year in which the event occurred |
| month | Integer representing the month in which the event occurred |
| day | Integer representing the day of the month in which the event occurred |
| DayOfYear | Day of the year in which the event occurred |
| decimalLongitude | Geographic longitude in decimal degrees |
| decimalLatitude | Geographic latitude in decimal degrees |
| depth | Depth in meters at which the event occurred |
| individualCount | Number or enumeration value representing the quantity of organisms |
| Comments | Additional comments or notes about the record |
| datasetID | Identifier for the dataset |
| datasetName | Name identifying the dataset from which the record is derived |
| basisOfRecord | Nature of the record, based on the Darwin Core terms |
| sourceArchive | Source from which the records were obtained |
| institutionCode | Name or acronym of the institution having custody of the object or information referred to in the record |
| seqnum | Identifier for the grid cell |
| lme_name | Name of the Large Marine Ecosystem (LME) region |
| abbrev_lme_name | Abbreviation of the Large Marine Ecosystem (LME) region name |
| verbatimScientificName | Original scientific name recorded |

| | |
|---|---|
| ModifiedName | Taxonomic name after modification or correction to improve accuracy or consistency |
| ReduceName | Taxonomic name after reduction to a higher taxonomic rank when the original name contained multispecies or complex designations |
| parse.name | Taxonomic name used for verification with AlgaeBase and/or WoRMS, obtained through parsing and formatting processes to ensure compatibility and consistency with the databases |
| openNomenclature | Uncertainty or provisional status of taxonomic identification |
| acceptedNameUsageID | Identifier for the current accepted scientific name details |
| scientificName | Scientific name according to AlgaeBase and/or WoRMS |
| scientificNameID | Identifier for scientific name details |
| URI | Set of identifiers constructed according to the generic syntax for Uniform Resource Identifiers |
| acceptedNameUsageAuthorship | The authorship information for the acceptedNameUsage formatted according to the conventions of the applicable nomenclatural Code |
| taxonomicStatus | Status of the use of the acceptedNameUsage as a label for a taxon |
| nomenclaturalStatus | Status related to the original publication of the name and its conformance to the relevant rules of nomenclature |
| taxonRank | Taxonomic rank of the most specific name in the acceptedNameUsage |
| taxonRankID | Identifier for the taxonomic rank of the most specific name in the acceptedNameUsage |
| kingdom | Full scientific name of the kingdom in which the taxon is classified |
| phylum | Full scientific name of the phylum or division in which the taxon is classified |
| class | Full scientific name of the class in which the taxon is classified |
| order | Full scientific name of the order in which the taxon is classified |
| family | Full scientific name of the family in which the taxon is classified |
| genus | Full scientific name of the genus in which the taxon is classified |
| isMarine | Boolean flag indicating whether the taxon is a marine organism, i.e., can be found in/above sea water |
| isBrackish | Boolean flag indicating whether the taxon is an organism that can be found in brackish water |
| isFreshwater | Boolean flag indicating whether the taxon occurs in freshwater habitats, i.e., can be found in/above rivers or lakes |

| | |
|---|---|
| isTerrestrial | Boolean flag indicating the taxon is a terrestrial organism, i.e., occurs on land as opposed to the sea |
| isFossil | Boolean flag indicating whether the taxon is an extinct organism |
| isToxic | Boolean flag indicating whether the taxon is a toxic organism |
| isHarmful | Boolean flag indicating whether the taxon is a harmful organism |
| bibliographicCitation | Bibliographic reference for the resource |
| bibliographicCitation_ToxicHarmfulStatus | Bibliographic reference for the resource's toxic and harmful status |
| modified | Date on which the resource was changed |
| database | Database source used for scientific name verification |

## 2.4 List of potentially toxic and/or harmful algal species

In the context of this study, "HA" is used as an abbreviation to refer collectively to potentially toxic and/or harmful algal species. Our comprehensive global list of HA species consists of the IOC-UNESCO taxonomic reference list (http://www.marinespecies.org/hab/; last accessed: June 20, 2023; Lundholm et al., 2009). We subsequently supplemented this list by incorporating the taxa list of Bates et al. (2020, 2019), which notably included the Ciliophora, *Mesodinium rubrum* (Lohmann) Leegard. We chose to retain *M. rubrum* due to its significant ecological implications (McKenzie et al., 2020; Olsen et al., 2019). We excluded the dinoflagellate *Protoperidinium crassipes* (Kofoid) Balech from Bates et al. (2020), since this heterotrophic species appears to act more as a toxin vector than a toxin producer (e.g., Tillmann et al., 2009). Each taxon was verified as described in the previous section and merged based on the "acceptedNameUsage" column. We assigned the toxic and/or harmful status to each record, following the criteria of Lundholm et al. (2009) and Bates et al. (2020, 2019). This compiled list includes 113 Dinoflagellata, 49 Heterokontophyta, 43 Cyanobacteriota, 11 Haptophyta, and one Ciliophora species. Of these, 205 species have been identified as toxic (indicated by a flag in the "isToxic" column; Table 1), seven are considered harmful (indicated by a flag in the "isHarmful" column; Table 1), and five species remain under debate regarding their toxic and/or harmful status (flagged in both "isToxic" and "isHarmful" columns; Table 1).

## 2.5 Data merger and synthesis

The filters implemented during the data merging and synthesis process aimed to ensure the quality and relevance of the dataset. The filters applied were as follows:

- Records without year information were removed to ensure data quality and enable meaningful temporal analysis, as the absence of this crucial temporal component would limit the dataset's usability for studying time-dependent patterns or trends.
- Records with depths greater than 2500 meters were excluded, considering the specific characteristics and depth ranges of the Arctic region based on bathymetry data.

▪ Records classified as "fossil only" or "fossil" in either the AlgaeBase or WoRMS databases (e.g., "isFossil" column;
Table 1) were excluded to focus only on currently occurring microbial plankton species. However, records classified
as freshwater or brackish according to the AlgaeBase or WoRMS databases (i.e., "isFreshwater" and "isBrackish"
columns; Table 1) were retained to account for their ecological relevance and potential responses to changing Arctic
conditions, given the Arctic's connection to freshwater and brackish coastal regions.
▪ Records not found in either AlgaeBase or WoRMS were excluded to ensure the inclusion of taxonomically validated
and accepted names.
▪ Taxa belonging to specific kingdoms (i.e., Animalia, Fungi, Acritarcha), phyla (i.e., Foraminifera, Oomycote,
Rhodophyta, Retaria), and classes (i.e., Phaeophyceae, Ulvophyceae) were excluded to maintain the focus on
microbial plankton species.
▪ Records identified at a taxonomic level higher than genus were removed from the dataset to ensure consistent and
accurate taxonomic classification at the genus level. Retaining records at the genus level allows for a more detailed
understanding of the composition of microbial plankton communities in the study area.
▪ Duplicate records were removed, using the following columns: "day", "month", "year", "depth", "decimalLatitude",
"decimalLongitude", "verbatimScientificName", "scientificName", "acceptedNameUsage", "basisOfRecord" and
"individualCount" (Table 1). This step ensured that each unique sampling event was represented by a single record
in the dataset.

After applying these filters, the dataset contains 385 348 individual georeferenced data points and 18 268 unique sampling
events (Fig. 1A; Schiffrine et al., 2024). To access the comprehensive diversity of HA species, we further subset the database
based on the "isToxic" and "isHarmful" columns (Table 1), resulting in a dataset with a total of 48 555 georeferenced data
points of HA species and 6744 unique sampling events (Fig. 1B; Schiffrine et al., 2024).
**2.7 Data analysis**
The size of each LME region was determined by calculating the total number of grid cells (i.e., $n_{total}$). For each LME region
and each month, the number of grid cells containing records was counted and summed per year (i.e., $n_{sampled}$). This value was
then divided by $n_{total}$ to estimate the percentage of the region that was sampled, or sample coverage, that specific year within
each LME region. The same method was applied to the HA sub-dataset, where $n_{HA\ sampled}$ represents the number of grid cells
containing HA records summed per year. Mapping and statistical analysis were performed on the filtered dataset (see section
2.6) in R (R version 4.4.1; Team and R Development Core Team, 2019), using *tidyverse* (version 2.0.0; Wickham et al., 2019),
*ggOceanMaps* (version 2.2.0; https://mikkovihtakari.github.io/ggOceanMaps/; Vihtakari, 2021), *vegan* (version 2.6-6.1;
Oksanen et al., 2020), and *betapart* (version 1.6; Baselga and Orme, 2012) packages.

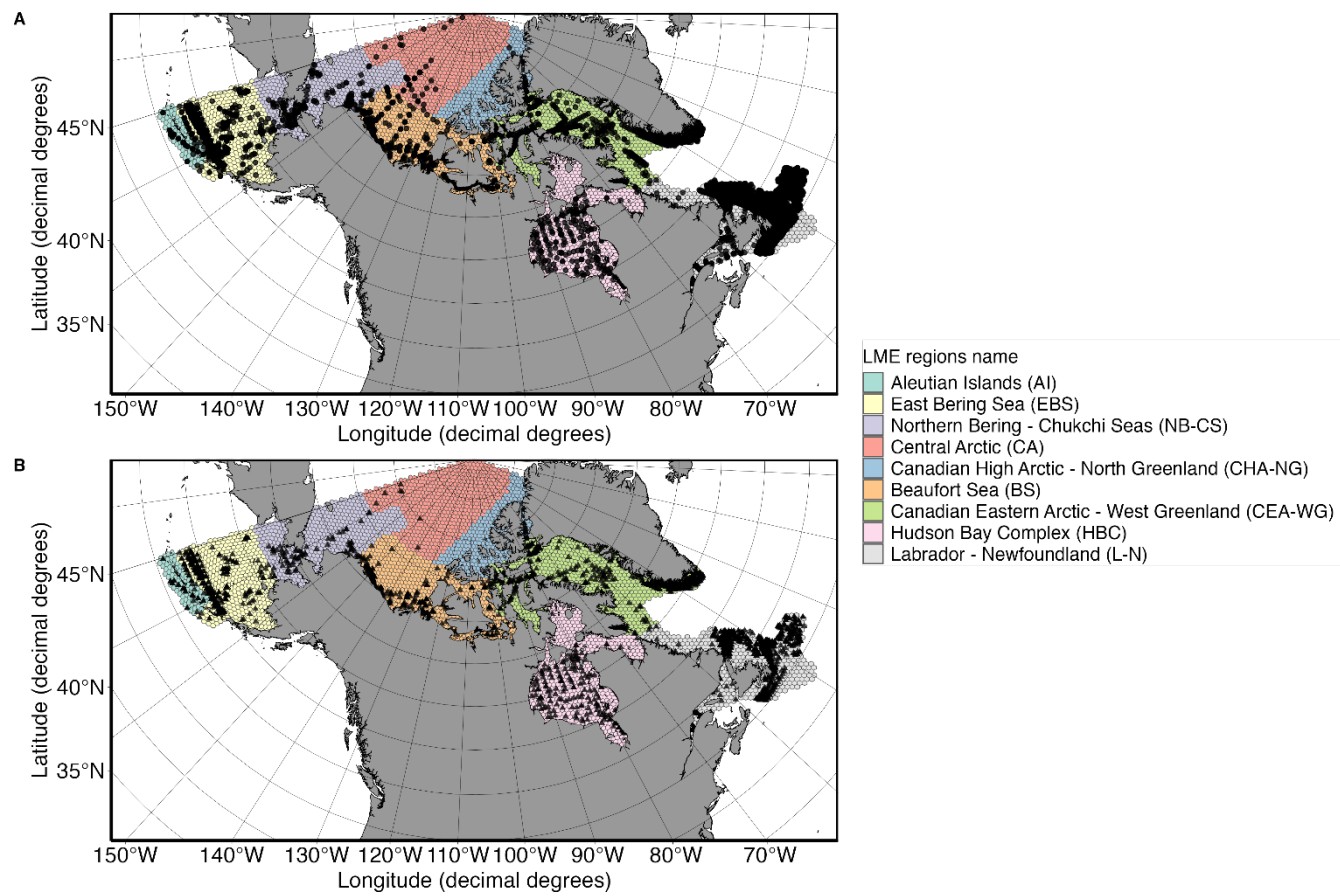


Figure 1: (A) Global distribution of microbial plankton occurrence and (B) potentially toxic and/or harmful algal (HA) species
records. Abbreviations of the Large Marine Ecosystems (LME) regions are in parentheses.

## 3 Results and discussion

### 3.1 Spatiotemporal coverage

The use of long-term datasets has significantly improved our understanding of microbial plankton taxa distribution and
diversity, as well as the underlying drivers of these patterns at both local (McKenzie et al., 2020; Nohe et al., 2020) and global
scales (Benedetti et al., 2021; Hallegraeff et al., 2021; Righetti et al., 2019). However, currently available databases on
microbial plankton occurrences only provide limited information on the Arctic Ocean. Although PhytoBase (i.e., Righetti et
al., 2020) is one of the most comprehensive and up-to-date sources of information on phytoplankton occurrence, data above
60° N are generally underrepresented. Furthermore, the recent study by Hallegraeff et al. (2021) did not specifically address
the evolution of HA blooms in the Arctic Ocean, but instead grouped the North American region of the Arctic Ocean within
the broader region of "East Coast America".

Despite the existence of several published microbial plankton taxa lists specifically focused on the North American sector of
the Arctic Ocean, there is currently a lack of a comprehensive and freely available standardized database accessible to the
scientific community. To fill this gap, our project aimed to compile and integrate a large and diverse collection of data from
multiple sources. The objective was to create a comprehensive database covering the distribution of microbial plankton
including photoautotrophic prokaryotes and eukaryotes (i.e., phytoplankton), as well as heterotrophic, phagotrophic, and
mixotrophic protistan species with a particular focus on HA species across the North American sector of the Arctic Ocean.
Our efforts greatly expanded the spatial and temporal coverage of microbial plankton data across all LME regions in this sector
of the Arctic Ocean compared to PhytoBase. Our database covers an impressive time span of 132 years, from 1888 to 2020,
with 95% of the data collected after 1963. Sampling was mainly concentrated between the months of June to September,
corresponding to reduced ice cover in the Arctic Ocean and better accessibility by ships to the Arctic. The spatial distribution
of the records was highly unbalanced, with 82% of data records falling in the Labrador—Newfoundland region alone, followed
by the Canadian Eastern Arctic—West Greenland with 8% (Figure S1). The remaining regions contribute smaller proportions,
ranging from 0.1% to 5% of the data records (Figure S1). The dataset spans a depth range from 0 to 1010 meters. However,
17% of the dataset lacks depth information, necessitating caution when interpreting the vertical distribution of phytoplankton
and other protists. This is particularly significant in the Arctic marine environment, where subsurface chlorophyll *a* maxima
are common (e.g., Martin et al., 2012). The scarcity of depth data may lead to an underestimation of biodiversity in these
deeper layers. While 83% of the entries include depth information, allowing for some general statements regarding vertical
distribution, the vast majority of data focuses on surface layers (95%), and the gaps in depth records impose certain limitations
on our ecological interpretations. Regarding the types of records within the dataset (i.e., basisOfRecord column, Table 1), the
majority (71%) are categorized under "HumanObservation", which includes occurrences documented through field notes,
literature, or records without physical or machine-recorded evidence. Another significant portion, 19% of the dataset falls
under the "PreservedSpecimen" category, representing samples that have been treated with fixatives for preservation.

**3.2 Taxonomic coverage**

A total of 1442 unique taxa were recorded in our study. This number falls within the range reported by Archambault et al.
(2010) and Poulin et al. (2011) for the same region (i.e., 1657 and 1229 taxa, respectively). It's essential to acknowledge that
both Archambault et al. (2010) and Poulin et al. (2011) conducted their analyses based on literature reviews predominantly
reliant on microscopic observations. In more recent comprehensive pan-Arctic taxonomic inventories using genomic
techniques, Lovejoy et al. (2017) and Ibarbalz et al. (2023) reported 2241 and 3082 different OTU taxa, respectively. The
discrepancy in reported taxa between our study and the aforementioned studies can be attributed to the fundamental differences
in our respective approaches—our reliance on mainly microscopic observations (i.e., >90%) versus their exclusive use of
genomic data. Genomic techniques possess the capacity to identify a broader spectrum of species, including those of smaller
size or less conspicuous under microscopic examination, such as the Mamiellophyceae *Micromonas polaris* Simon, Foulon &
Marin. Microscopic observations, which constitute a substantial portion of our dataset, inherently introduce certain biases.
They may overlook rare or small species (<3 μm) and encounter challenges related to precise species identification,
compounded by considerations such as the choice of fixative (e.g., acidic Lugol's solution or formalin; Sournia, 1978). As a
result, our study may not offer a fully comprehensive representation of total species richness, particularly concerning rare or
molecularly detectable taxa.

In this study, Heterokontophyta and Dinoflagellata were the most commonly occurring phyla, accounting for approximately
45% and 36% of total occurrences, respectively (Fig. 2A). Within the phylum Heterokontophyta, which notably included
diatoms (Guiry et al., 2023), the genus *Chaetoceros* Ehrenberg was the most frequently observed, followed by *Thalassiosira*
Cleve, which accounted for 24% and 14% of total Heterokontophyta occurrences, respectively (Fig. 2B). *Tripos* Bory and
*Gyrodinium* Kofoid & Swezy were the two most abundant genera in the phylum Dinoflagellata, accounting for 20% and 16%
of total Dinoflagellata occurrences, respectively (Fig. 2C). The observed predominance of Heterokontophyta in this study,
particularly the genera *Chaetoceros* and *Thalassiosira,* is in line with the general understanding of Arctic phytoplankton and
other protist diversity (Lovejoy et al., 2017; Poulin et al., 2011). On the other hand, the findings for phylum Dinoflagellata
contrast with prior research that has highlighted the predominance of the genus *Protoperidinium* Bergh (Okolodkov and
Dodge, 1996). The exceptionally high occurrence of *Tripos* and *Gyrodinium* should be interpreted with caution. These two
genera are mainly observed in the Labrador—Newfoundland region, where the majority of the data collected originates from
the Continuous Plankton Recorder (CPR; Figure S2). It is important to note that CPR uses a large mesh size (270 μm) (e.g.,
Richardson et al., 2006), resulting in an over-representation of larger taxa, such as *Tripos* and *Gyrodinium*.

Cyanobacteria are notably scarce in Arctic waters, and their ecological roles appear to be taken over by picoeukaryotic algae
(Buitenhuis et al., 2012; Pedrós-Alió et al., 2015). Studies indicate a marked decrease in the cell abundance of oceanic
picocyanobacteria with increasing latitude in the northern and southern hemispheres. Among these, *Prochlorococcus*
Chisholm, Frankel, Goericke, Olson, Palenik, Waterbury, West-Johnsrud & Zettler ex Komárek et al., the most abundant
photosynthetic genus in tropical oceans, is notably absent from polar waters (Flombaum et al., 2013). The International Census
of Marine Microbes (ICoMM) surveys retrieved fewer than 30 true Cyanobacterial tags overall, reflecting their scarcity in
Arctic marine waters (Lovejoy et al., 2011). In addition, a study of molecular diversity in the Beaufort Sea identified
picocyanobacteria that were mostly affiliated with freshwater and brackish *Synechococcus* Nägeli lineages rather than oceanic
ones (Waleron et al., 2007). Similarly, a 16S rRNA gene study of bacterial communities in the Beaufort Sea and Amundsen
Gulf found that cyanobacteria were not among the top 50 most abundant bacterial taxa (Comeau et al., 2011). Our study
revealed the presence of 26 distinct cyanobacteria entries, with 12 identified at the genus level and 14 at the species level.
These 26 taxa encompass a diverse range of Cyanobacteria, with *Synechococcus* ~~Nägeli~~ being the most frequently detected
genus, accounting for 91% of the cyanobacterial occurrences. Overall, cyanobacterial occurrences represent a small fraction
of the total (0.8%). These results are consistent with the hypothesis of an allochthonous origin of Cyanobacteria in the coastal
Arctic Ocean, as all species observed in our study are labeled by AlgaeBase as freshwater species (Schiffrine et al., 2024). Our
findings also support previous studies suggesting that the presence of Cyanobacteria in the Arctic may be underestimated and
highlight the need for further research to understand their ecological significance.

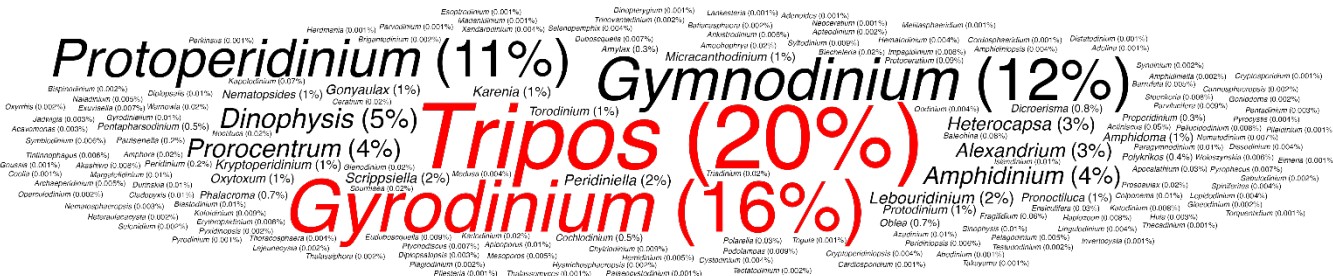

**Figure 2: Word cloud representation of the (A) most occurring phyla, (B) most occurring genera within the phylum Heterokontophyta, and (C) most occurring genera within the phylum Dinoflagellata. Words are sized proportionally to their occurrence and colored to distinguish the two most frequent categories.**


A well-established concept is the latitudinal gradient of diversity, where the highest levels of diversity are typically found near
the equator, gradually diminishing towards the poles (Ibarbalz et al., 2019; Righetti et al., 2019). However, this pattern is not
universally consistent across all taxa. Chaudhary et al. (2016) has highlighted that for certain groups, diversity may exhibit a
bimodal distribution, with peaks occurring in the mid-latitudes rather than at the equator, and a notable decline in species
richness within equatorial regions. Furthermore, patterns of phytoplankton diversity exhibit significant variation between
different Longhurst provinces (Hörstmann et al., 2024). The primary driver behind these patterns is likely ocean temperature
variability (Chaudhary et al., 2017; Ibarbalz et al., 2019). While environmental conditions undoubtedly contribute to these
diversity patterns, the scarcity of data may also account for the observed low diversity. Righetti et al. (2020) reported a total
of 1704 phytoplankton species, including 239 species within the same grid used in our study. However, our study detected an
additional 1359 taxa, of which 532 belonged to Heterokontophyta and 363 to Dinoflagellata. Our results indicate that previous
research may have significantly underestimated the biodiversity of Arctic phytoplankton and other protists (Righetti et al.,
2019). Such underestimation may bias our understanding of the latitudinal gradient of diversity.
**3.3 Difference in species richness according to the Arctic LME regions**
LME regions show substantial variation in the nature of the data (i.e., basisOfRecords column; Table 1; Figure S3A) and data
provenance (i.e., sourceArchive column; Table 1; Figure S3A), resulting in high variation in sampling coverage (Figure S4).
Regions with higher contributions from OBIS/GBIF, such as the Labrador—Newfoundland region (Figure S3A), tend to have
more extensive datasets. This leads to greater sampling coverage (Figure S4) and a higher probability of capturing a wider
range of species, providing a more comprehensive representation of local biodiversity. In contrast, regions with higher
contributions from "Individual Studies", such as the Aleutian Islands, Central Arctic, or Canadian High Arctic—North
Greenland regions (Figure S3A), may have been the focus of more specific scientific research. This results in lower sampling
coverage (Figure S4) and may underrepresent species richness, potentially leading to an incomplete understanding of the true
species composition in these areas. To address this concern, we used the Chao2 index, a widely used nonparametric method
for estimating species richness in a community (Chao and Shen, 2003). The Chao2 index is particularly valuable as it accounts
for rare species, providing a more accurate estimate of species richness in datasets with uneven sampling effort. The application
of the Chao2 index allows us to assess alpha diversity, i.e., diversity on a local scale, within each LME region, especially when
working with frequency counts or presence/absence data. The Chao2 index reveals significant differences in diversity among
the LME regions (Fig. 3). The Canadian Eastern Arctic—West Greenland and Beaufort Sea regions exhibited the highest
values of the Chao2 index, indicating higher species richness and diversity within their phytoplankton and other protist
communities. In contrast, the Aleutian Islands and Central Arctic regions had the lowest values, suggesting lower species
richness and diversity in these areas. The East Bering Sea, Hudson Bay Complex, Labrador—Newfoundland, and Northern
Bering—Chukchi Seas regions also showed moderate to high Chao2 index values, indicating varying levels of species richness
and diversity across these regions.

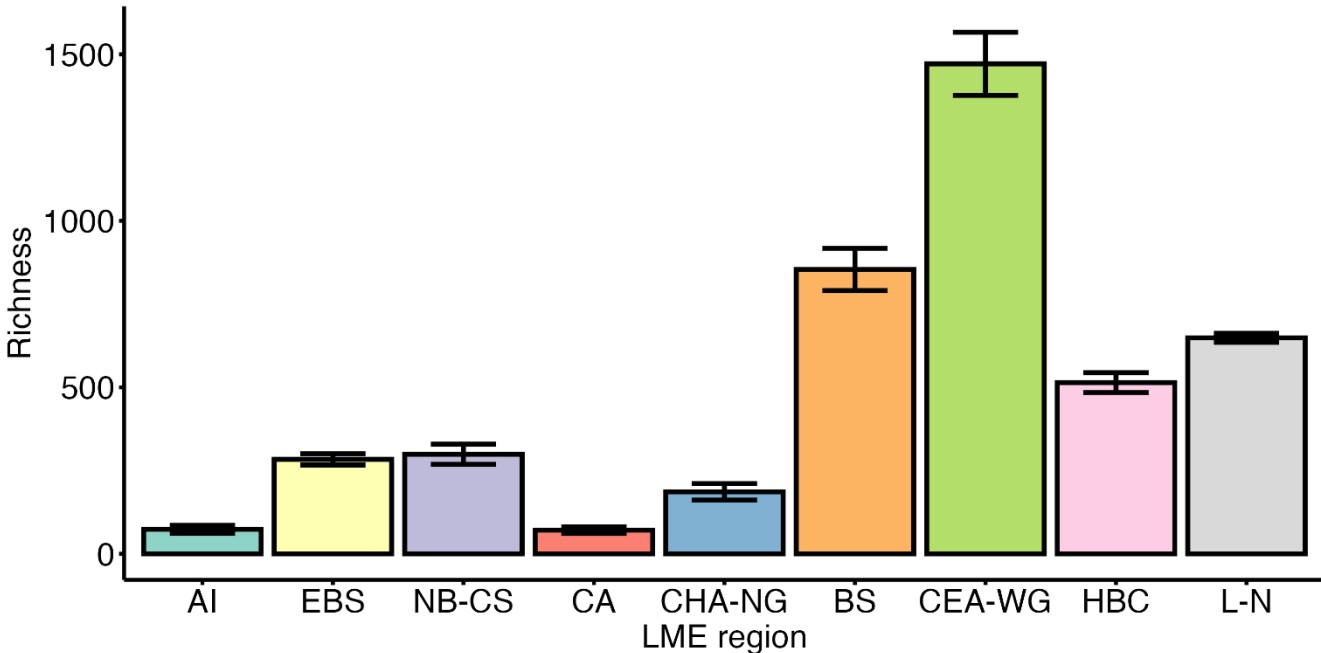


**Figure 3: Chao2 index for each Large Marine Ecosystem (LME) region. Error bars represent the standard error (i.e., SE). LME regions are labeled as follows: AI (Aleutian Islands), n = 28; EBS (East Bering Sea), n = 401; NB-CS (Northern Bering—Chukchi Seas), n = 122; BS (Beaufort Sea), n = 318; CA (Central Arctic), n = 29; CHA-NG (Canadian High Arctic—North Greenland), n = 19; CEA-WG (Canadian Eastern Arctic—West Greenland), n = 656; HBC (Hudson Bay Complex), n = 177; L-N (Labrador—Newfoundland), n = 7268.**

316

To further analyze local diversity, we used species accumulation curves (SACs) to illustrate the number of species sampled relative to the level of sampling effort (Thompson and Withers, 2003). SACs typically reach an asymptote when sufficient sampling effort is achieved, enabling us to estimate the comprehensiveness of species richness detection. In our analysis, we computed SACs based on the number of species observed in each grid cell for each month of every year, yielding valuable insights into species richness and the extent of saturation (i.e., completeness of species richness detection) between regions (Fig. 4). The Hudson Bay Complex and Labrador—Newfoundland regions exhibited saturation at cumulative richness levels of around 400 and 600 taxa, respectively (Fig. 4), indicating that a significant proportion of the taxa present in these regions had been sampled. Conversely, the SACs for other regions did not reach a plateau, suggesting that the sampling effort was insufficient to capture the complete diversity (Fig. 4). This pattern was particularly pronounced in the northernmost regions, such as Central Arctic and Canadian High Arctic—North Greenland (Fig. 4).

327

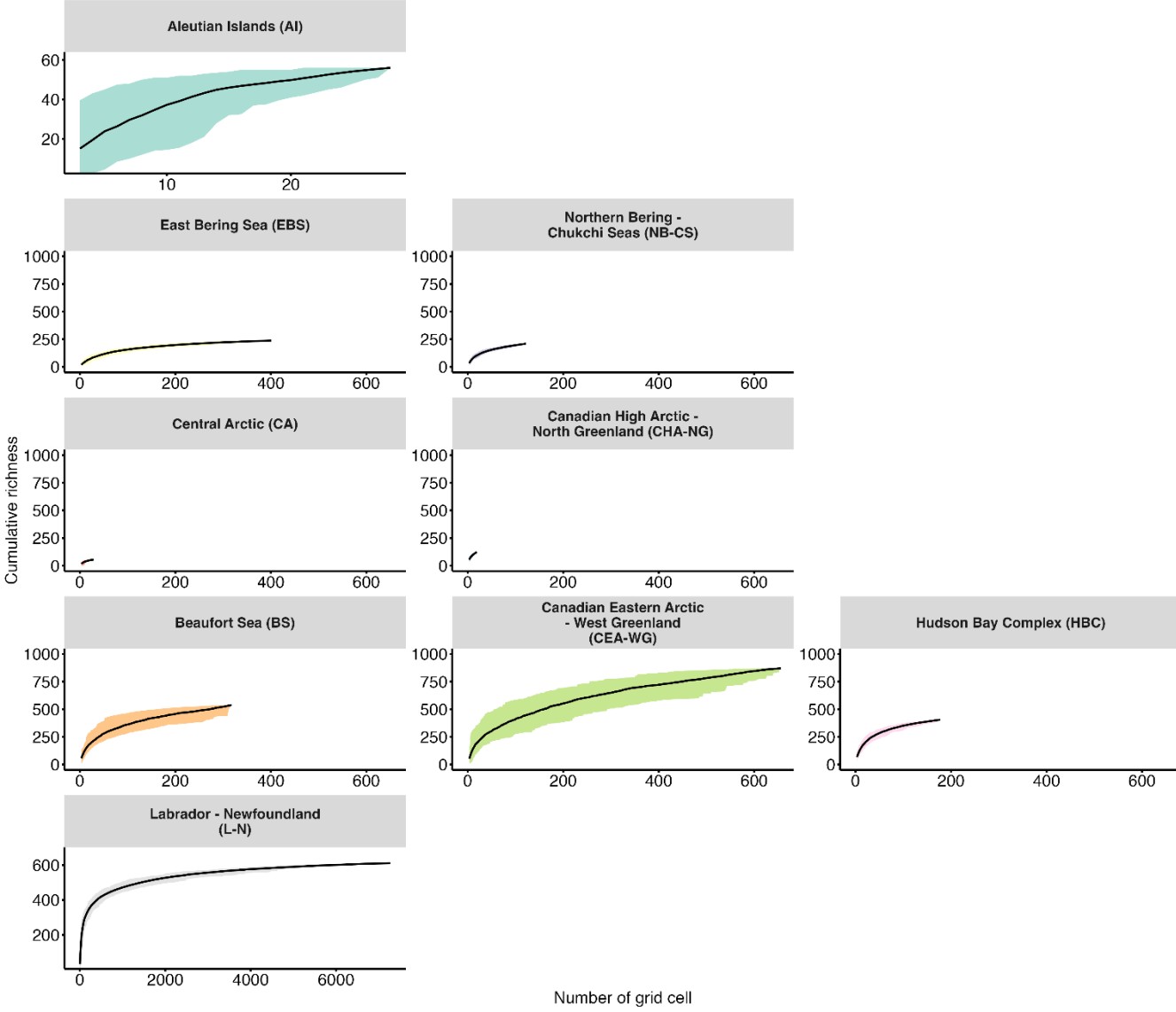

329

**Figure 4: Cumulative richness as a function of number of grid cell for each month of every year with standard deviation (i.e., SD; colored area) for each Large Marine Ecosystem (LME) region. LME region abbreviations are in parentheses.**

332

The Central Arctic region, known for its extreme environmental conditions, such as low-nutrient concentrations and prolonged annual sea-ice cover (Codispoti et al., 2013), exhibits a lower Chao2 index compared to other regions (Fig. 3). However, the SAC did not saturate (Fig. 4), indicating that the actual diversity in this region may be higher than observed in this study. Despite the inflow of nutrient-rich water through the Bering Strait (Torres-Valdés et al., 2013), which contributes to the high productivity in the Pacific regions, such as East Bering Sea and Northern Bering—Chukchi Seas (Tremblay et al., 2015), these

regions display a relatively low Chao2 index (Fig. 3). One possible explanation for this observation is that the sampling effort conducted in these regions may not have been sufficient to capture the complete range of species diversity, leading to an underestimation of richness. This is supported by the SACs (Fig. 4), which show that the curves for both regions do not reach a plateau, indicating that the sampling effort was insufficient to fully assess the diversity present in these areas. The Labrador—Newfoundland region displays an intermediate Chao2 index (Fig. 3), although the SAC indicates that the majority of species have been recorded (Fig. 4). This saturation can be attributed to the provenance (i.e., "sourceArchive" column; Table 1; Figure S3A) and the nature (i.e., "basisOfRecords" column; Table 1; Figure S3B) of the data. Notably, 90% of the data come from OBIS, the majority of which were collected using the Continuous Plankton Recorder (CPR). The CPR's sampling methods, which involve a large mesh size (270 µm), fixed sampling depth (5–10 m), and high sampling speed (15–20 knots), are effective at capturing larger and more robust species but tend to miss smaller and more delicate species, as well as those that are not consistently present in the surface mixed layer (Richardson et al., 2006). As a result, the intermediate Chao2 index and the apparent saturation in the SACs may coexist. This is because the limitations of the CPR result in under-sampling of certain species, leading to apparent saturation in the species accumulation curves, while the Chao2 index may suggest that more species are present but not captured by the current sampling methods. The Chao2 index values for both the Beaufort Sea and the Canadian Eastern Arctic—West Greenland regions were remarkably high (Fig. 3). This is consistent with previous observations of relatively high diversity in the Canadian Eastern Arctic—West Greenland region (Joli et al., 2018; Kalenitchenko et al., 2019). However, it comes as a surprising result for the Beaufort Sea region, where, except during episodic upwelling events, the water column is highly stratified, and nutrient concentrations in the surface mixed layer are extremely low, especially in the northern part (Ardyna et al., 2017). This typically leads to a community with relatively low diversity and a high prevalence of picoeukaryotes, mostly represented by the psychrophilic prasinophyte *Micromonas polaris* N.Simon, Foulon & B.Marin (Balzano et al., 2012; Coupel et al., 2015; Tremblay et al., 2009). The relatively high Chao2 index value for the Beaufort Sea region may be explained by the fact that most of the samples were collected from nearshore areas (Figure S5). These nearshore areas are known for their high productivity (Ardyna et al., 2017), likely due to their exposure to nutrient-rich waters that can support the growth and diversity of microbial plankton communities. Another explanation could be the influence of stable surface mixed layer in the Beaufort Gyre that provide multiple ecological niches, further contributing to the observed high diversity. Nevertheless, the SACs for both regions indicate that sampling efforts in these areas are incomplete (Fig. 4). This implies that diversity may be underestimated and underscores the importance of further sampling to gain a more accurate understanding of local biodiversity in both the Beaufort Sea and the Canadian Eastern Arctic—West Greenland regions.

The beta diversity (β) assessment provides valuable insights into the dissimilarity of taxa composition between multiple samples, enabling researchers to understand the variation in biodiversity across different spatial scales (Whittaker, 1972). In this study, we used the Sørensen dissimilarity index ($\beta_{SØR}$) as the β diversity index to determine the proportion of taxa that are not shared between LME regions. The $\beta_{SØR}$ values range from 0 to 1, where 0 indicates identical taxonomic composition at all

sites, and 1 represents completely different sets of taxa (Baselga, 2010). Our analysis revealed the subdivision of the LME regions into three distinct clusters based on their taxa composition (Fig. 5). The first cluster, known as the "Pacific Cluster", includes the Aleutian Islands, East Bering Sea, and Northern Bering—Chukchi Seas regions. The second cluster, referred to as the "Northern Arctic Cluster", encompasses the Central Arctic and Canadian High Arctic—North Greenland regions. Lastly, the third cluster, named the "Mixed Arctic Cluster", consists of the Beaufort Sea, Canadian Eastern Arctic—West Greenland, Hudson Bay Complex, and Labrador—Newfoundland regions. The grouping of the Aleutian Islands, East Bering Sea, and Northern Bering—Chukchi Seas regions is anticipated owing to their common water supply and circulation patterns, which involve receiving water inflows from the Pacific Ocean through the Bering Strait (Rudels and Carmack, 2022). Consequently, this leads to comparable environmental conditions and nutrient inputs, which, in turn, explain the observed similarities in the composition of microbial plankton communities. Similarly, the Central Arctic and Canadian High Arctic—North Greenland regions share a common water circulation pattern in the Arctic Ocean (Rudels and Carmack, 2022). The common circulation pattern, combined with similar environmental characteristics, contributes to the similarities in taxonomic composition between these two regions. The inclusion of the Beaufort Sea region with Atlantic-dominant regions (i.e., Canadian Eastern Arctic—West Greenland, Hudson Bay Complex, and Labrador—Newfoundland) into one unique cluster may initially seem contradictory due to its geographic location outside the Atlantic side of the Arctic and lack of direct influence from Atlantic waters (Rudels and Carmack, 2022). However, this clustering is based on similarities in taxa composition rather than geographic proximity or environmental conditions. Despite its location, the Beaufort Sea region exhibits a higher resemblance in taxa composition to the Canadian Eastern Arctic—West Greenland, Hudson Bay Complex, and Labrador—Newfoundland regions compared to other regions in the dataset. This unexpected similarity may be attributed to oceanic circulation patterns and water mass transport mechanisms that connect these regions (Rudels and Carmack, 2022). These circulation patterns and transport mechanisms may facilitate the dispersal of taxa from the Beaufort Sea to Atlantic-dominated regions, thereby influencing the observed similarities in taxa composition (Wassmann et al., 2015). Such circulation-driven dispersal was documented by Reid et al. (2007), who observed the spread of the diatom *Neodenticula seminae* (Simonsen & T.Kanaya) Akiba & Yanagisawa from the Northwest Arctic to the Atlantic side, possibly through the Canadian Arctic Archipelago and/or Fram Strait. This observation provides additional support for the concept of shared microbial plankton composition influenced by oceanic circulation. These findings provide important insights into the biogeographic patterns of microbial plankton communities in the Arctic LME regions and highlight the importance of considering both geographic and ecological factors when interpreting these patterns.

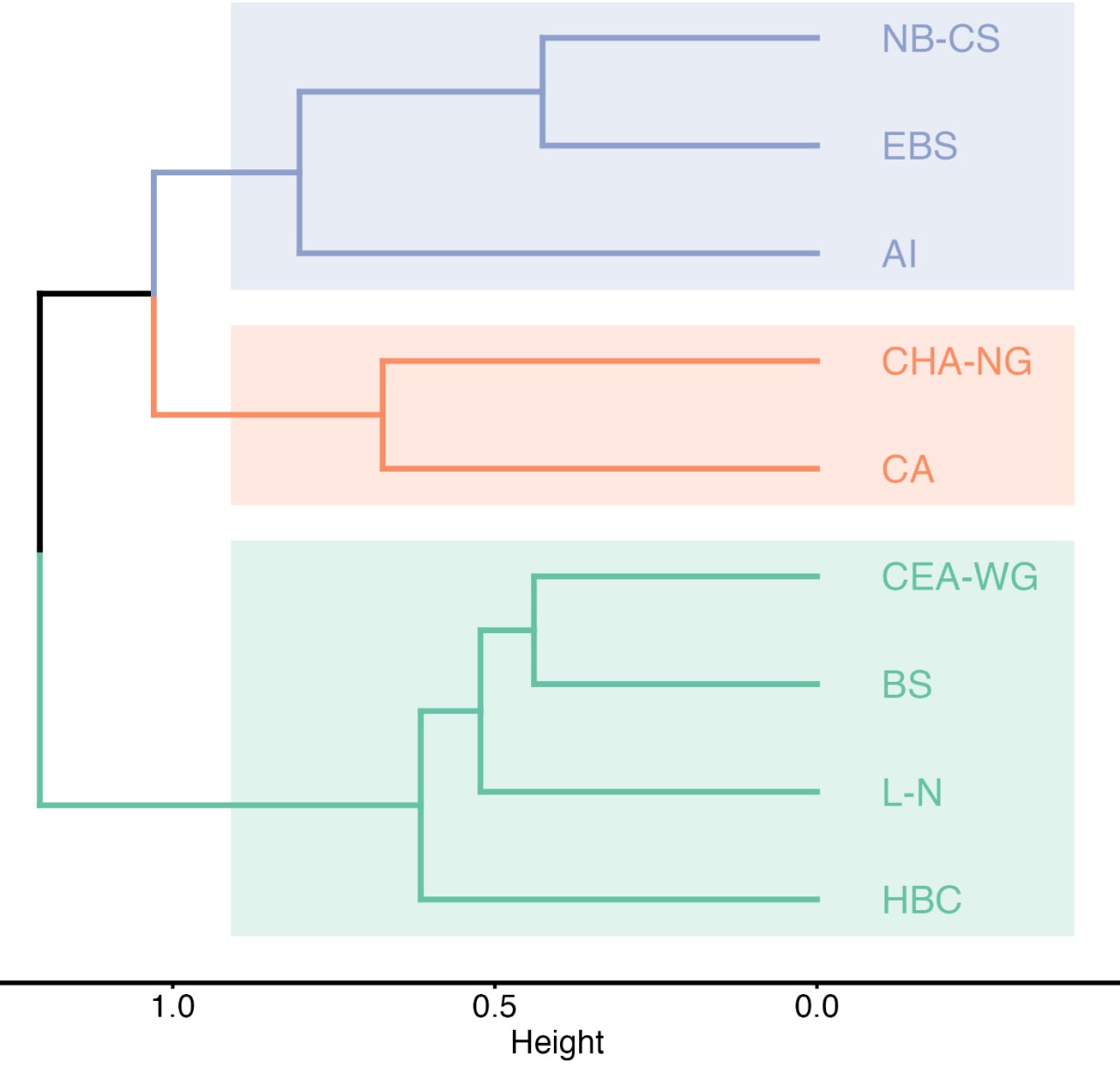

401

**Figure 5: Cluster analysis of β diversity Sørensen dissimilarity index (βSØR) between the different Large Marine Ecosystem (LME) region obtained with Ward's cluster method. LME regions are labeled as follows: NB-CS (Northern Bering—Chukchi Seas), EBS (East Bering Sea), AI (Aleutian Islands), CHA-NG (Canadian High Arctic—North Greenland), CA (Central Arctic), CEA-WG (Canadian Eastern Arctic—West Greenland), BS (Beaufort Sea), L-N (Labrador—Newfoundland) and HBC (Hudson Bay Complex).**

407

## 3.4 Diversity of potentially toxic and/or harmful algal species

The presence of HA species has been a well-known concern in temperate marine and freshwater ecosystems, but their occurrence in the Arctic marine ecosystem is relatively new. With the ongoing climate change in the Arctic Ocean, there is a high probability that the frequency of HA occurrence will increase, notably by stimulating cyst germination (Anderson et al., 2021). Furthermore, the expansion of HA distributions from other regions due to increased ship traffic in the Arctic may further exacerbate this problem (e.g., Chan et al., 2019). HA species pose substantial risks to both human and ecosystem health, and can cause massive economic losses through fish kills. The phycotoxins produced by some of these organisms can bioaccumulate in higher trophic level organisms, including molluscs, seabirds, and marine mammals. When transferred to higher trophic levels, these phycotoxins can result in massive mortality, neurological or gastrointestinal adverse effects if consumed at concentrations that surpass safe thresholds. In the Alaskan sector of the Bering Sea, concentrations of these toxins have been detected in shellfish tissues that could pose a health risk to local populations (Gao et al., 2019). This issue is particularly important, as northern populations rely on traditional harvesting of fish, shellfish, and marine mammals for subsistence food.

Of the 217 HA species compiled from Lundholm et al. (2009) and Bates et al. (2020, 2019) (see section 2.4 for details; Schiffrine et al., 2024), our database identified 59 species. Notably, our study detected a higher number of HA species compared to previous studies conducted by Poulin et al. (2011) and Pućko et al. (2019), who reported 36 and 27 species, respectively, after updating their species lists with revised taxonomy. It is noteworthy that both studies primarily aggregated data from literature reviews based on microscopic observations. Our study contributed an additional 25 species, including 16 species from the phylum Dinoflagellata and seven species from the phylum Heterokontophyta. It is important to note that at least 11 species reported by Poulin et al. (2011) were not detected in the present work, as they occur in other Arctic regions not covered in our study, such as *Alexandrium minutum* Halim observed in the Russian and Scandinavian regions.

Many of the species highlighted in our study are of particular concern for the Arctic Ocean due to their production of phycotoxins. Based on Lundholm et al. (2009) and Bates et al. (2020, 2019), we identified 48 potentially toxin-producing species, as indicated by the "isToxic" flag (Table 1). Of the 73 accepted species included in the genus *Pseudo-nitzschia* (AlgaeBase. World-wide electronic publication, 2023; last access June 2024), 28 are known to produce domoic acid (Bates et al., 2019; Lundholm et al., 2009), with nine of these toxin-producing species being present in our database (Schiffrine et al., 2024). Meanwhile, at least 16 of the 45 accepted species in the genus *Alexandrium* are known to be toxic (AlgaeBase. World-wide electronic publication, 2023; last access June 2024; Lundholm et al., 2009), but only five species have been recorded in our database (Schiffrine et al., 2024). The dinoflagellate genus *Dinophysis* has 276 phototrophic and heterotrophic accepted species worldwide (AlgaeBase. World-wide electronic publication, 2023; last access June 2024), and 10 of these species have been found to produce various toxins (Lundholm et al., 2009). Additionally, 14 out of the 133 accepted species of the genus

*Prorocentrum* Ehrenberg have been confirmed to produce a range of toxins (AlgaeBase. World-wide electronic publication,
2023; last access June 2024; Lundholm et al., 2009). Our database contains at least six and five species from the genera
*Dinophysis* and *Prorocentrum*, respectively (Schiffrine et al., 2024).

Surprisingly, we observed the presence of *Pyrodinium bahamense* Plate in our dataset, an occurrence published by MGnify
(https://www.ebi.ac.uk/metagenomics; e.g., Mitchell et al., 2020) and hosted on GBIF
([https://www.gbif.org/dataset/b42d7c7f-43e5-4e24-abd7-fab3b4fceb09](https://www.gbif.org/dataset/b42d7c7f-43e5-4e24-abd7-fab3b4fceb09)). This observation is intriguing, as *P. bahamense* is
typically associated with warm tropical waters (Morquecho, 2019), and its presence in Arctic waters is unexpected. The
publication referenced by MGnify (e.g., Joli et al., 2018) does not mention the presence of *P. bahamense*, a fact also confirmed
by the authors (pers. comm.), further adding to the uncertainty of this occurrence. This discrepancy suggests potential
misidentifications when using data from platforms such as MGnify, which are primarily designed for microbiome analysis and
may not always accurately identify species. Similarly, we observed the presence of the Pelagophyceae *Aureococcus*
*anophagefferens* Hargraves & Sieburth, in the Canadian Eastern Arctic—West Greenland region. Unlike the occurrence of *P.*
*bahamense*, this finding was published by Elferink et al. (2017b) as a supplement to their work (e.g., Elferink et al., 2017a).
Therefore, this occurrence has undergone a peer-review process, reducing the likelihood of misidentification and increasing
the credibility of the occurrence. Additionally, the presence of this species in the Arctic region was also reported by Ibarbalz
et al. (2023), further supporting its presence in this region. Nevertheless, retaining such data is standard practice in scientific
research to ensure transparency and data integrity. These occurrences underscore the critical importance of rigorous validation
and verification processes in biodiversity studies, particularly when addressing unexpected findings in unique and sensitive
environments like the Arctic. Ensuring the accuracy of species identification through meticulous peer-review and cross-
referencing with established databases is essential. Such occurrences, if confirmed, could indicate significant ecological shifts
driven by climate change, emphasizing the necessity for continuous, comprehensive monitoring to accurately assess and
understand the broader impacts on species distributions and ecosystem dynamics.

Approximately 50% of all HA occurrences are primarily represented by only five species: *Pseudo-nitzschia delicatissima*
(Cleve) Heiden, *P. seriata* (Cleve) H.Peragallo, *Dinophysis acuminata* Claparède & Lachmann, *Prorocentrum cordatum*
(Ostenfeld) Dodge, and *Mesodinium rubrum* (Lohmann) Leegard (Figure S5). Among these species, *P. delicatissima*, *P.*
*seriata*, *P. cordatum*, and *M. rubrum* showcase an extensive geographic distribution, aligning with their broad prevalence
across different regions, including the Arctic (Figure S6) (Bates et al., 2020, 2018; Lassus et al., 2016). In contrast, *D.*
*acuminata* demonstrates a more confined geographic range (Figure S6). While the presence of *M. rubrum*, which serves as
prey for *D. acuminata* (Reguera et al., 2012), could potentially influence the distribution of this species, it cannot provide a
comprehensive explanation for the constrained range of *D. acuminata*. This notion is underscored by the relatively broader
geographic distribution observed for *M. rubrum* (Figure S6). Temperature is also unlikely to be another restricting factor, as
*D. acuminata* demonstrates tolerance to a temperature range from 4 to 10 °C, which corresponds to the temperatures observed

475 in the region where this species occurred. The observed limited distribution of *D. acuminata* may therefore arise from a

476 complex interplay of ecological and environmental factors that collectively shape its spatial pattern, a certainty that remains

477 elusive in the scope of the present study. Additionally, this limited distribution could also be attributed to insufficient studies

478 focused on this particular species.

479

480 Climate change is expected to cause HA species to shift towards northern latitudes, increasing their prevalence in the North

481 American Arctic region. In the Fram Strait, Nöthig et al. (2015) reported a dominance shift towards the harmful

482 prymnesiophyte species *Phaeocystis pouchetii*, likely driven by a warm water anomaly in the Atlantic waters of the West

483 Spitsbergen Current. Additionally, during a research cruise in the summer of 2022, an unprecedented bloom of *Alexandrium*

484 *catenella* (Whedon & Kofoid) Balech, in terms of its scale, abundance, and toxicity, was tracked as it moved through the

485 Bering Strait (Fachon et al., 2024). However, the extent of this northward progression of HA species in other Arctic regions,

486 particularly the North American sector, remains relatively unexplored. To address this gap, we examined the temporal variation

487 in the northernmost latitude where HA species are observed over the years. The analysis of maximum latitude of HA species

488 for each month and year (i.e., max. $\text{Lat}_{HA}$) reveals a gradual increase over time (Fig. 6A). To capture this trend more accurately,

489 we applied LOESS (Locally Estimated Scatterplot Smoothing) regression. LOESS is particularly suitable for our dataset,

490 which spans multiple decades and involves significant variability in sampling efforts. Unlike linear or polynomial regressions,

491 LOESS does not assume a fixed functional form. Instead, it fits localized regressions to small subsets of the data, resulting in

492 a smooth curve that better reflects the underlying patterns without imposing rigid assumptions about the data's structure. By

493 using LOESS, we can visualize trends in the data while accounting for the inherent variability, allowing for a clearer picture

494 of long-term patterns. The LOESS curve smooths out short-term fluctuations, revealing a marked increase towards the 90s

495 (Fig. 6A), emphasizing the accelerated northward progression of HA species in recent years. However, this trend is likely

496 influenced by heightened oceanographic research and expeditions in higher latitudes, as evidenced by the strong correlation

497 between max. $\text{Lat}_{HA}$ and the maximum recorded latitude (max. $\text{Lat}_{recorded}$; $\rho = 0.9$; p-value < 0.01; Fig. 6B). Nonetheless, this

498 association appears to exhibit variability depending on the species (Table 2). It is noteworthy that among the species analyzed,

499 there are 24 with insufficient available data to calculate the correlation (Table 2). For 12 species, there is a very weak Spearman

500 rank correlation ($-0.2 < \rho < 0.2$; Table 2), indicating no meaningful linkage between max. $\text{Lat}_{HA}$ and max. $\text{Lat}_{recorded}$. One such

501 example is the dinoflagellate species *Karenia mikimotoi* (Miyake & Kominami ex Oda) Hansen & Moestrup, which

502 consistently maintains a near-constant max. $\text{Lat}_{HA}$ despite increasing max. $\text{Lat}_{recorded}$ (Fig. 6C). This pattern suggests that while

503 the sampling efforts expand northward, *K. mikimotoi* seems to be restricted to a specific latitude range. This limited latitudinal

504 distribution is possibly attributed to its temperature tolerance range (4–30°C) (Li et al., 2019). The colder temperatures in the

505 North American Arctic align with the lower thermal limit of this species, likely acting as a thermal barrier to the dispersal of

506 *K. mikimotoi*. Conversely, 12 species demonstrate a strong positive correlation ($\rho > 0.6$; Table 2) that emphasizes a significant

507 relationship between max. $\text{Lat}_{HA}$ and max. $\text{Lat}_{recorded}$. For instance, the max. $\text{Lat}_{HA}$ of the raphidophyte species *Heterosigma*

508 *akashiwo* (Hada) Hada ex Hara & Chihara appears to be closely linked to max. $\text{Lat}_{recorded}$ (Fig. 6C; Table 2). This suggests the

possibility of *H. akashiwo* being a permanent resident of the North American Arctic algal community. However, there remains
uncertainty about whether the species observed in the database correspond to those found in temperate regions, as records of
*H. akashiwo* in our database are identified with qualifiers such as "cf." or "aff." (Bérard-Therriault et al., 1999), indicating
some uncertainty in their identification. In addition, Arctic conditions may not be conducive to its growth (Edvardsen and
Imai, 2006; Mehdizadeh Allaf, 2023). In particular, toxin production is lowest at 30 °C, and blooms of *H. akashiwo* have been
observed at temperatures ≥ 15 °C (Edvardsen and Imai, 2006; Mehdizadeh Allaf, 2023), suggesting that toxin production in
the Arctic might be significantly reduced due to lower temperatures. The findings concerning the constrained latitudinal
distribution of *K. mikimotoi* and the potential permanent residency of *H. akashiwo* in North American Arctic waters highlight
the significance of investigating environmental factors and biological traits that shape the distribution and abundance of HA
species Arctic Ocean and adjacent seas. In particular, gaining insights into thermal limits, growth requirements, and toxin
production of these species can provide valuable information on their responses to the evolving Arctic climate and potential
risks to human health and ecosystems. Further research is needed to investigate the population dynamics and ecological roles
of these HA species within the Arctic context, as well as their interactions with other marine organisms and the physical
environment. In addition, utilizing molecular techniques to confirm the identity of these species would help clarify whether *H.*
*akashiwo* is indeed a permanent resident of the North American Arctic algal community or if the records are due to
misidentifications or persistent contamination.

While studies have detected phycotoxins in the North American Arctic (Baggesen et al., 2012; Elferink et al., 2017a; Gao et
al., 2019; Hubbard et al., 2023; Li et al., 2016; Pućko et al., 2023), there are no reports of HA events at high latitudes (>60° N)
in the Harmful Algal Event Database (HAEDAT; http://haedat.iode.org/index.php; last accessed: October 2023). HAEDAT's
criteria for a HA event are strict, including toxin accumulation in seafood above safe levels, discoloration or scum in the water
causing ecosystem or socioeconomic damage, negative effects on humans, animals, or other organisms, or precautionary
closures of harvesting areas based on predefined thresholds of toxic phytoplankton cells in the water. This suggests that these
events may not meet the HAEDAT criteria and raises questions about reconsidering and potentially revising the criteria used
to monitor harmful algal blooms (HABs) in this unique region, particularly in the context of rapid environmental changes. The
impacts of HABs in the Arctic may be more subtle, chronic, and ecosystem-specific, and therefore might not trigger the
HAEDAT thresholds designed for more temperate regions. For instance, the Arctic marine ecosystem is particularly sensitive
to changes, and even low-level toxin presence can have significant impacts on local wildlife and indigenous communities that
rely heavily on marine resources. To address these unique challenges, it may be important to develop Arctic-specific
monitoring criteria that consider the distinct ecological and socioeconomic context of the region. Such criteria could include
lower thresholds for toxin levels, more sensitive indicators of ecosystem health, and an emphasis on the cumulative effects of
low-level exposures over time. Additionally, incorporating traditional ecological knowledge from indigenous communities
could enhance monitoring efforts, as these communities have long-standing observations and insights into local environmental
changes. Encouraging the adoption of these tailored criteria would support more comprehensive monitoring efforts, ensuring
that all relevant harmful algal events are documented. This would not only improve our understanding of HABs in the Arctic
but also inform better management and mitigation strategies to protect the Arctic marine ecosystem and the communities that
depend on it.

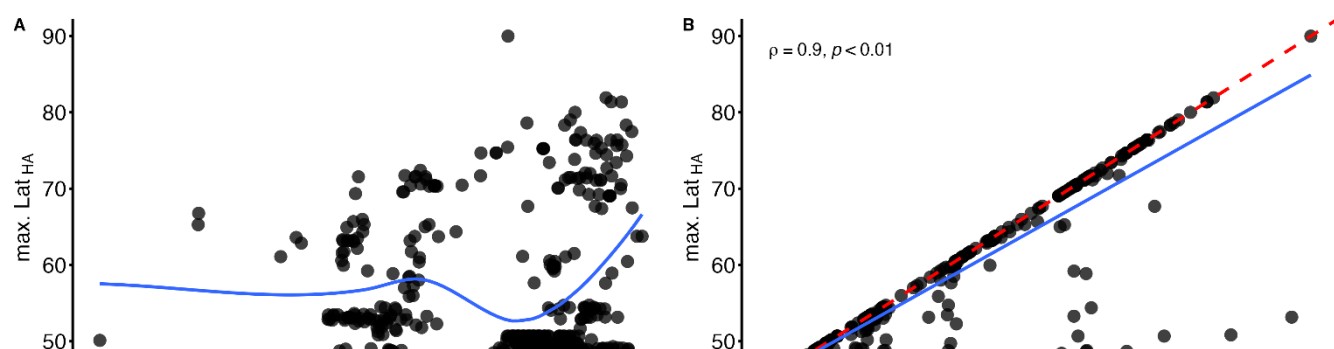

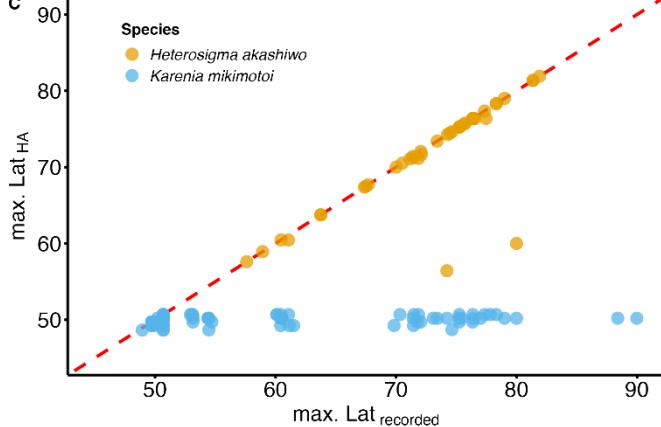

**Figure 6: (A)** Temporal variation of the maximum latitude of HA species for each month and year (max. Lat$_{HA}$); the blue line
represents the locally estimated scatterplot smoothing (LOESS). **(B)** Relationship between maximum latitude of HA species for each
month and year (max. Lat$_{HA}$) and the maximum latitude recorded for each month and year (max. Lat$_{recorded}$); the blue line represents
the linear model; the red dashed line represents the 1:1 slope. **(C)** Relationship between the maximum latitude of HA species for
each month and year (max. Lat$_{HA}$) and the maximum latitude recorded for each month and year (max. Lat$_{recorded}$) for *Heterosigma
akashiwo* (yellow circle) and *Karenia mikimotoi* (blue circle); the red dashed line represents the 1:1 slope.

## 4 Data availability

The dataset described in this work is published in the Zenodo repository: https://zenodo.org/records/13376814 (Schiffrine et
al., 2024).

## 5 Code availability

The code used in this study is publicly accessible on Zenodo https://zenodo.org/records/13376814 (Schiffrine et al., 2024). This repository contains the scripts and tools used for various aspects of our study, including data conversion, data quality control, analysis, and visualization.

**Table 2: Summary of the Spearman rank correlation ($\rho$) analysis between the maximum latitude of HA species for each month and year (max. $Lat_{HA}$) and the maximum latitude recorded for each month and year (max. $Lat_{recorded}$) for each HA taxon.**

| Phylum | Class | Species | $\rho$ | p-value |
|---|---|---|---|---|
| **Ciliophora** | Litostomatea | *Mesodinium rubrum* Lohmann | 0.381 | *** |
| **Cyanobacteria** | Cyanophyceae | *Dolichospermum spiroides* (Klebahn) Wacklin, L.Hoffmann & Komárek | — | — |
| | | *Planktothrix agardhii* (Gomont) Anagnostidis & Komárek | — | — |
| **Dinoflagellata** | Dinophyceae | *Alexandrium catenella* (Whedon & Kofoid) Balech | — | — |
| | | *Alexandrium monilatum* (J.F.Howell) Balech | — | — |
| | | *Alexandrium ostenfeldii* (Paulsen) Balech & Tangen | -0.042 | NS |
| | | *Alexandrium pseudogonyaulax* (Biecheler) Horiguchi ex K.Yuki & Y.Fukuyo | -0.177 | NS |
| | | *Alexandrium tamarense* (Lebour) Balech | 0.026 | NS |
| | | *Amphidinium carterae* Hulburt | -0.097 | NS |
| | | *Amphidinium klebsii* Kofoid & Swezy | — | — |
| | | *Amphidinium operculatum* Claparède & Lachmann | — | — |
| | | *Dinophysis acuminata* Claparède & Lachmann | 0.322 | *** |
| | | *Dinophysis acuta* Ehrenberg | 0.46 | *** |
| | | *Dinophysis norvegica* Claparède & Lachmann | 0.144 | * |
| | | *Dinophysis ovum* F.Schütt | — | — |
| | | *Dinophysis tripos* Gourret | -0.7 | * |
| | | *Gonyaulax spinifera* (Claparède & Lachmann) Diesing | 0.078 | NS |
| | | *Gymnodinium catenatum* H.W.Graham | — | — |
| | | *Karenia mikimotoi* (Miyake & Kominami ex Oda) Gert Hansen & Moestrup | 0.164 | ** |
| | | *Lingulodinium polyedra* (F.Stein) J.D.Dodge | — | — |
| | | *Margalefidinium fulvescens* (M.Iwataki, H.Kawami & Matsuoka) F.Gómez, Richlen & D.M.Anderson | — | — |
| | | *Margalefidinium polykrikoides* (Margalef) F.Gómez, Richlen & D.M.Anderson | — | — |

| | | | | |
|---|---|---|---|---|
| | | *Phalacroma rotundatum* (Claparéde & Lachmann) Kofoid & J.R.Michener | 0.014 | NS |
| | | *Prorocentrum concavum* Y.Fukuyo | — | — |
| | | *Prorocentrum cordatum* (Ostenfeld) J.D.Dodge | 0.561 | *** |
| | | *Prorocentrum emarginatum* Y.Fukuyo | — | — |
| | | *Prorocentrum lima* (Ehrenberg) F.Stein | 0.038 | NS |
| | | *Prorocentrum mexicanum* Osorio-Tafall | — | — |
| | | *Prorocentrum micans* Ehrenberg | -0.536 | * |
| | | *Prorocentrum rhathymum* A.R.Loeblich III, Sherley & R.J.Schmidt | — | — |
| | | *Protoceratium reticulatum* (Claparède & Lachmann) Bütschli | 0.515 | *** |
| | | *Pyrodinium bahamense* L.Plate | — | — |
| | Noctilucophyceae | *Noctiluca scintillans* (Macartney) Kofoid & Swezy | — | — |
| | Syndiniophyceae | *Hematodinium* Chatton & Poisson | — | — |
| **Haptophyta** | Coccolithophyceae | *Chrysochromulina leadbeateri* Estep, Davis, Hargraves & Sieburth | — | — |
| | | *Haptolina ericina* (Parke & Manton) Edvardsen & Eikrem | 0.005 | NS |
| | | *Haptolina hirta* (Manton) Edvardsen & Eikrem | 0.867 | *** |
| | | *Phaeocystis pouchetii* (Hariot) Lagerheim | 0.638 | *** |
| | | *Prymnesium parvum* N.Carter | -0.489 | * |
| | | *Prymnesium polylepis* (Manton & Parke) Edvardsen, Eikrem & Probert | -0.014 | NS |
| | | *Pseudohaptolina birgeri* (Hällfors & Niemi) Ribeiro & Edvardsen | 0.8 | * |
| **Heterokontophyta** | Bacillariophyceae | *Pseudo-nitzschia australis* Frenguelli | — | — |
| | | *Pseudo-nitzschia delicatissima* (Cleve) Heiden | 0.681 | *** |
| | | *Pseudo-nitzschia fraudulenta* (Cleve) Hasle | — | — |
| | | *Pseudo-nitzschia granii* (Hasle) Hasle | — | — |
| | | *Pseudo-nitzschia obtusa* (Hasle) Hasle & Lundholm | 0.881 | *** |
| | | *Pseudo-nitzschia pseudodelicatissima* (Hasle) Hasle | 0.703 | *** |
| | | *Pseudo-nitzschia pungens* (Grunow ex Cleve) Hasle | 0.68 | *** |
| | | *Pseudo-nitzschia seriata* (Cleve) H.Peragallo | 0.695 | *** |
| | | *Pseudo-nitzschia turgidula* (Hustedt) Hasle | 0.803 | * |
| | Coscinodiscophyceae | *Corethron pennatum* (Grunow) Ostenfeld | 0.69 | *** |
| | Dictyochophyceae | *Dictyocha fibula* Ehrenberg | 0.688 | *** |
| | Dictyochophyceae | *Octactis speculum* (Ehrenberg) F.H.Chang, J.M.Grieve & J.E.Sutherland | 0.673 | *** |
| | Mediophyceae | *Chaetoceros concavicornis* Mangin | 0.4 | *** |
| | | *Chaetoceros convolutus* Castracane | 0.175 | ** |

| | | | |
|---|---|---|---|
| | *Chaetoceros debilis* Cleve | 0.387 | *** |
| | *Leptocylindrus minimus* Gran | 0.367 | *** |
| Pelagophyceae | *Aureococcus anophagefferens* Hargraves & Sieburth | — | — |
| Raphidophyceae | *Heterosigma akashiwo* (Hada) Hada ex Y.Hara & Chihara | 0.868 | *** |

*NS, p-value > 0.05; *, p-value < 0.05; **, p-value < 0.01; ***, p-value < 0.001; — indicates not tested.*

## 6 Conclusion

Several databases exist that document the occurrence of microbial plankton taxa in temperate marine ecosystems. However,
these resources often have limited representation of polar ecosystems or may lack such data entirely. Given the substantial
environmental changes in the Arctic Ocean and their impact on microbial plankton communities, it is crucial to expand our
understanding of Arctic phytoplankton biodiversity and biogeography. This study compiled various sources of digital
biological records, both published and unpublished, to create a comprehensive dataset for North American Arctic marine
microbial plankton occurrences. This dataset encompasses 385 348 individual georeferenced data points and 18 266 unique
sampling events, covering 1422 species, including key phyla like Heterokontophyta, Dinoflagellata, Haptophyta, Ciliophora,
and others. This effort addresses the historical limitations of Arctic microbial plankton data, which were often confined to
specific regions or lacked comprehensive geographic and date-referenced records (Poulin et al., 2011).
Our study provides the largest database to date on the occurrence of microbial plankton species, including photoautotrophic
prokaryotes and eukaryotes (i.e., phytoplankton), as well as heterotrophic, phagotrophic, and mixotrophic protistan species in
the North American part of the Arctic. This dataset can serve as a valuable resource for investigating the biogeography and
phenology of microbial plankton taxa in the region, particularly when integrated with other published datasets. Through the
application of geostatistical methods, our database contributes to a refined understanding of potential changes in Arctic
microbial plankton communities in the future. Additionally, by supplementing our dataset with information concerning the
toxicity or harmful nature of species, it facilitates assessments of the potential proliferation of toxic and/or harmful species
within the Arctic Ocean.
Moreover, in light of the comprehensive analysis conducted in this study, it becomes evident that routine phycotoxin
monitoring should encompass the North American Arctic. Historically, this region has been overlooked, driven by the
assumption that high-latitude Arctic areas are not prone to significant toxic algal blooms or phycotoxin contamination.
Nevertheless, our research, which sheds light on the diversity, distribution, and prevalence of HA species within this distinct
area, emphatically underscores the urgency to reassess this perspective. The conspicuous presence of HA species underscores
the imperative for a comprehensive and proactive monitoring strategy.

## Author contributions

NS led the study, the data processing and archiving, and the writing. FD, MP, SL, AR, and MG provided the quality-controlled data from different Arctic regions. KD initiated the collection of data in Iqaluit and surrounding areas of Frobisher Bay as part of the Baseline program in 2019. All authors reviewed and commented the manuscript.

## Competing interests

The authors declare that no competing interests are present.

## Acknowledgements

This work was supported by Fisheries and Oceans Canada (DFO) (Ocean Protection Plan and Ocean and Freshwater Science Contribution Program), Natural Sciences and Engineering Research Council of Canada, ArcticNet (Network of Centres of Excellence of Canada) and Fonds de recherche du Québec—Nature et technologies through the strategic cluster Québec-Océan to M. Gosselin and A. Rochon. We are grateful to local community members, DFO and university collaborators who participated in sampling and taxonomic analysis. We thank the AlgaeBase team for their invaluable assistance in providing us with an API key. This is a contribution to the research programs of the Institut des sciences de la mer of UQAR and Québec-Océan.

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
