# Peer review of "Microbial plankton occurrence database in the North American Arctic region: synthesis of recent diversity of potentially toxic and/or harmful algae"

_Earth System Science Data, 2024_

## Author Response (AR1)

We thank the referee for taking the time to review this manuscript and for his thorough and insightful comments. The feedback was invaluable in refining the clarity and depth of our study. We greatly appreciate the constructive criticism, which guided us in addressing key areas for improvement. In the following sections, we have provided detailed responses to each comment and made the necessary revisions to the manuscript to reflect the suggestions. We believe that these changes have strengthened the overall quality of the paper, and we are grateful for the referee's contribution to this process.

RC1 : Line 33: subjected is not the right word. This implies harm. The Arctic has been the focus of ..
**Response: We replaced the word "subjected to" with "the focus of" as suggested (L31).**

RC1: Line 46. the reference is over a decade old. chang past decade to recent decades..
**Response: We have shortened this paragraph to focus more on the issues directly relevant to our data (L37-45). The text mentioning the "past decade" has been deleted.**

RC1 Line 85. Agreed quantitative data are limited, but I do not see how the review here addresses this concern.
**Response: The sentence has been changed to: "The paucity of data on the richness and diversity of Arctic microbial species…" (L68). We have compiled results from several studies that provide insights into Arctic phytoplankton diversity. While these studies do not comprehensively cover all regions, they provide valuable snapshots of phytoplankton communities that can serve as a foundation for future research. In addition, the review provides methodological recommendations to improve future data collection efforts. These include the integration of molecular techniques with traditional microscopy, and the use of long-term monitoring programs to capture temporal variability. Finally, this is one of the most recent and comprehensive reviews of phytoplankton that aims to be exhaustive in terms of geographic scope and temporal coverage.**

RC1 : figure 1, I am not sure how the LMEs were delineated. I would question the inclusion of Bering Sea areas as these are not the Arctic per se. the the clear break would be Bering Strait where the water is entrained into the Arctic.
**Response: We understand the concern of the referee regarding the delimitation of LMEs in Figure 1. The boundaries of each LME region were defined using spatial data processed with the mregions package (https://github.com/ropensci/mregions; https://docs.ropensci.org/mregions/reference/mr_shp.html; https://docs.ropensci.org/mregions/reference/mr_names.html), which integrates authoritative marine spatial information from**

**https://www.marineregions.org/. This approach is documented in detail in the LME_Arctic.R script provided in our Zenodo repository (Schiffrine et al., 2024;https://zenodo.org/records/13376814) and outlined in Section 2.2 of the paper (L99, 104-107). The LMEs were delineated according to established guidelines to ensure consistency and accuracy in defining these ecologically significant regions. To address the concern about the inclusion of Bering Sea areas, we have added a line in the response to clarify our rationale (L104-107). We acknowledge that the inclusion of these areas may seem counterintuitive. However, the delineation followed the guidance provided by the mregions package, which takes into account the physical and ecological characteristics of marine regions. The decision to include Bering Sea areas within the Arctic LMEs reflects their inclusion in broader ecological and oceanographic contexts, especially considering the transport dynamics through the Bering Strait that entrain water into the Arctic region. It's important to note that we imposed only specific east-west boundaries during the delineation process. No further adjustments were made to the delineated boundaries beyond these imposed constraints.**

RC1 : Line 228:  the June to September is consistent with accessibility by ships to the Arctic, not seasonal dyanamics, which would correspond to ice free or light available periods. (e.g.April to october).
**Response: The sentence has been rephrased to be more explicit (L211-212).**

RC1: Given the paucity of depth data and information it is not possible to make general statements.  This is in essence a report of species with little or no  possibility of.  ecological interpretation
**Response: The sentences have been rephrased for clarity (L215 - 221). Here, only 17% of the data lack depth sampling information, which means that 83% of the entries contain depth information. It's not impossible to make general statements about vertical distribution, but certain limitations must be taken into account.**

RC1: Given this is a catalog why are "molecular" only records not included in this reveiw?
**Response: This study relies primarily on web-based search engines, online database queries, and data from the ArcticNet campaign, which encompasses a wide array of sources, including molecular data. It is important to note that our work is primarily a synthesis of available data, rather than a single publication of "our data" reflecting what is currently available. Thus, the inclusion of exclusively "molecular" records in this review was not within the scope of our methodology, which focused on compiling comprehensive data accessible through multiple sources.**

RC1 : This is the crux of why the richness and diversity comparisons between regions is missleading at best. The regions were not sampled using the same techniques.  The CPR is restricted to Large cells in surface waters.
**Response: We agree with the referee's observation and acknowledge the limitations of comparing richness and diversity between regions sampled using different techniques. Again, this study relies primarily on web-based search engines, online database queries, and data from ArcticNet, and we do not impose specific limitations on the nature of the data. The purpose of this paragraph was to highlight the overrepresentation of some genera due to the specific characteristics of the CPR sampling method. Given these constraints, we have employed the Chao2 index in our subsequent analyses. Our goal with this study is to provide a comprehensive overview of phytoplankton diversity in the North American Arctic, despite the inherent methodological limitations.**

RC1 : L273 No one has ever denied that Synechococcus occurs in the Bering Sea, or even Atlantic water entering the Arctic.  What was stated and obvious in previous publications is that these do not persist once in  the Arctic Basin.  The coastal reports are all of Freshwater to maybe brackish  (sal of 5) species of cyanos.   I would not call the data given her a diverse range of cyanobacteria.
**Response: Indeed, it is not necessarily their presence that is generally underestimated, but rather their importance and/or abundance. The entire paragraph has been rewritten to be more explicit (L256-272).**

RC1: delete "conventional"
**Response: This term has been removed (L288-291).**

RC1: Given the non-comparative data set. Why present a Choao2 diversity index. Or why only give Chao2?
**Response: As the referee pointed out, and as explained in lines 293-305, we used the Chao2 index primarily because of the nature of our data, and secondly because of the inherent imbalance of the dataset across regions, which increases the likelihood of missing rare species. The Chao2 index, based on incidences in sample sites, is a widely recognized nonparametric method for estimating species richness in a community. This method is particularly appropriate for our data as it addresses the issue of incomplete sampling by providing an estimate of the minimum species richness based on the frequency of species occurrences. Furthermore, since our data are based on occurrence and not abundance, abundance-based estimates such as Chao1, ACE, Simpson, and Shannon indices are not applicable. These indices rely on individual counts within species, which are not available in our dataset. Therefore, the choice of the Chao2 index is justified and appropriate for our study, ensuring that our biodiversity estimates are as accurate as possible given the constraints of our data.**

However, as the referee pointed out, it is important to compare different indicators. Therefore, we have included various species richness indices in the R code (L445 in Arctic_dataBaseline_Article.R; note that we are now using the vegan package to calculate these various indices). The indices included are: Bootstrap ± SE (boot); Chao2 ± SE (Chao2); First-order Jackknife ± SE (jack1); Second-order Jackknife (jack2); Species count (Species), as illustrated in the figure below.

[Figure]

We observe that the various indicators are relatively consistent across most regions, except for the BS and CEA-WG regions, where the Bootstrap and First-order Jackknife estimates are both lower than the Chao2 and Second-order Jackknife estimates. This discrepancy can be attributed to the sensitivity of Chao2 and Second-order Jackknife to rare species and uneven sampling effort. These indices adjust for the presence of species that occur in only one or two samples, leading to higher estimates in regions with many rare species. In contrast, the Bootstrap and First-order Jackknife methods may not fully account for these rare species, resulting in lower estimates.

RC1: Figure 3 Plot all but e Aleutian Islands  and Labrador -Newfoundland curves  as a single figure. this puts the data from the 7 Arctic areas on the same scale and would make comparisons in the context of sampling effort and show how total species numbers varied between regions. none are close  to flattening out except the L-N plot, which is a totally biased data setl, as stated in  line 341.
**Response: The panels have been rearranged and the scales for the 7 "Arctic" regions have been adjusted to be consistent (both x- and y-axes) as suggested by the referee (see revised Fig. 4). This adjustment allows for a better comparison of the species accumulation curves between regions, making it easier to assess the impact of sampling effort and to observe variation in the total species number between regions.**

RC1: As I cannot seem to access the data, I cannot verify, but I think, the nearness to shore is not a reason for high diversity in the Beafort. It could be the stable layers

in the beaufort gyre providing multiple niches,  More riverine flow from the MacKensie or a host of other reasons.

**Response: Data is available on Zenodo https://zenodo.org/records/13376814; see section 4 and 5. We have updated the text to address this concern (L362-365).**

RC1: Line 347, Is there an a priori reason that productivity and diversity should be correlated in the sea. After all HA blooms are highly productive and fix carbon at high rates, but are not diverse species assemblages.

**Response: We have updated the text to address this concern (L354-365).**

RC1: Line 413 …  Using an entire paragraph on a single occurrence wit a reference to a dead internet link is not justifiable.  A more rational explanation is that it was a misidentification or a carryover contamination from a phytoplankton net. One sentence is all that is needed.

**Response: The paragraph is indeed is a bit long. Our intention was to emphasize the importance of caution when using genomic data from platforms such as MGnify, which can sometimes lead to misidentifications. Given the increasing reliance on genomic techniques and automated analysis tools, it is important to highlight potential problems and the need for thorough verification of species identifications.**

**We agree that a more concise mention of this event is appropriate. The paragraph has been modified to keep the focus on the broader findings, while noting the importance of data accuracy and verification. In addition, we have included an explanation of the *Aureococcus* occurrence in the text alongside the *Pyrodinium* occurrence (L445-463).**

RC1: Figure 5A: I miss the point of the polynomial regression fitting.

**Response: In Figure 5A, we applied a LOESS regression to our dataset to identify and visualize trends over time in the maximum latitude of HA occurrences. LOESS regression is particularly useful for capturing complex patterns and trends without assuming a specific functional form. Given the variability and non-linear trends in our dataset, which spans multiple decades and sampling efforts, LOESS provides an accurate representation of the underlying patterns. In addition, LOESS regression provides a smoothed curve that helps visualize the central tendency of the data over time, revealing trends that may be obscured by short-term fluctuations. By applying LOESS regression, we aimed to provide a clearer picture of how the maximum latitude of HA occurrences has changed over time, allowing us to observe potential shifts in the HA distribution.**

RC1: Fig 5 b teh colors on the graph do not agree with the colors for Heterosigma and Karenia given in the legend. The distribution of r looks lie a result of baseline misidentification or persistent contamination.

**We have changed the color in the figure to match the legend. The distribution of *Heterosigma akashiwo* (Hada) Hada ex Hara & Chihara indeed shows some intriguing patterns, and we acknowledge the potential issues of misidentification, as we indicated in the text (L517-519).**

RC1: Table 2. the Dinoflagellates phylum should be showen next to Alexandrium catenella
Suggest Writing out the subphylum name for Diatoms next to Chaetoceros concavicornis. Also suggest Aureococcus needs as much or more explanation than Pyro.. Was this a missidentification as well?

**Response: We have added a class column in the revised Table 2. We have also added an explanation of the *Aureococcus* occurrence in the text next to the *Pyrodinium* occurrence "flag" (L452-463).**

We thank the referee for taking the time to review this manuscript and for his thorough and insightful comments.

RC2: My main critique is that the authors use the term "phytoplankton" "for simplicity" (L. 13/14). Phytoplankton is a defined term (see ESA data ontologies: https://data.esa.int/esado/en/) and, therefore, misused in this study's context. Making this shortcut will not help users understand the content of this database or how to use it correctly. I recommend the authors summarize their data based on the higher taxonomic ranks, such as the kingdom level (Chromista, Cyanobacteriota, Eubacteria, Protozoa, Plantae, and unassigned Eukaryota).
Unassigned Eukaryota includes several entries of 'Medusa,' which is neither microbial plankton nor phytoplankton. I would recommend removing these entries from further analysis
**Response: We agree that this simplification may be misleading. We deleted the sentence (see L11-13). In the manuscript, when appropriate, we used the term "Microbial planktonic." Regarding the entries of 'Medusa,' this was primarily an error in the scientific name quality control step within the wm_record() function. We have since corrected this error (see Taxonomy_dataBaseline_Arctic.csv in Schiffrine et al. 2024; https://zenodo.org/records/13376814).**

RC2: L21: The diversity presented is not unexpected, particularly in pan-Arctic genomic surveys (e.g., Ibarbalz et al. 2023 https://doi.org/10.1525/elementa.2022.00060 reporting >3000 OTUs). The novelty of this dataset is certainly the richness of observational taxonomic counts.\
**Response: We agree with the comment. We removed the "revealing greater diversity than previously thought" from the text (L17-19).**

RC2: L.26 I think it's more widely accepted that HA occurs in the Arctic, including a few HAB events.
**Response: We revised our statement to reflect this broader understanding while still emphasizing the importance of our findings and the need for extensive and long-term sampling efforts (L23-28).**

RC2: L.39-54 This paragraph is irrelevant in the context of the presented data
**Response: We partially agree with this observation. We believe that providing a physico-chemical context for the Arctic is important to frame our study. However, this paragraph has been shortened (L37-42).**

RC2: L.59/60 As suggested above, I recommend removing the term "phytoplankton." The authors did not do any trophic assignment of species, which allowed them to discuss the different trophic groups in more detail.
**Response: This sentence has been deleted.**

RC2: L.79/80 Nöthig et al. do not introduce the concept of Atlantification. Please explain what you mean by this term.
**Response: The sentence has been changed. We now refer to the warm anomaly event in eastern Fram Strait to explain the shift towards to harmful prymnesiophytes (L61-63).**

RC2: L.98 Please explain what you mean by "the largest database of its kind" or remove it.
**Response: Indeed, this statement was somewhat presumptuous. The sentences have been changed (L79-84).**

RC2: For transparency, including the number of hits of the respective keyword searches in OBPS, PANGAEA, and GBIF would have been interesting.
**Response: The number of data has been included for each source (L91, 93, 95).**

RC2: L.207-209 Please provide the version of the packages used.
**Response: The version of each package used has been indicated (L100, 104, 185, 186, 187).**

RC2: L.228/229 does not necessarily coincide with the seasonal dynamics but with accessibility to sampling sites due to ice cover.
**Response: The sentence has been rephrased to be more explicit (L215-221).**

RC2: L.357-269 A Figure would have been helpful to see the different abundances of taxa between regions
**Response: In the revised version of the document, we have included a word cloud representation (see new Figure 2) to visualize the most common phyla and genera within the Heterokontophyta and Dinoflagellata phyla. In addition, Supplementary Figure 2 shows the occurrence of *Gyrodinium* and *Tripos* within each Large Marine Ecosystem (LME) region. Together, these new figures provide a clear visual representation of the occurrence of the taxa in each LME region.**

RC2: L273, I suggest making differences on the genus level. How can you be sure that there are 27 distinct taxa? For example, Aphanothece spp. and Aphanothece clathrate could essentially be the same species.
**Response: The number of entries within the phylum Cyanobacteria was counted (i.e., section "## 3.1 Taxonomic coverage: Nb genus Cyanobacteria" in Arctic_dataBaseline_Article.R; L1371; https://zenodo.org/records/13376814). This includes as many genera as species. We have reviewed our counts to clarify the distinctiveness of the taxa identified in our study. We have updated the text to reflect the exact number of genera and species identified (L265-266).**

RC2: L.278/279 I don't think cyanobacteria's presence is generally underestimated. Several studies on bacterial abundances in the Arctic, see, e.g., Ortega-Retuerta 2012 (https://doi.org/10.5194/bg-9-3679-2012), where bacteria contributed >80% to the local PP.
**Response: Indeed, it is not necessarily their presence that is generally underestimated, but rather their importance and/or abundance. The entire paragraph has been rewritten to be more explicit (L255-271).**

RC2: L.282-284 Studies have also shown that diversity patterns are bimodal (Chaudhary et al. 2016; https://doi.org/10.1016/j.tree.2016.06.001) and can vary between Longhurst provinces (Hörstmann et al. 2021; https://doi.org/10.1111/1462-2920.15832).
**Response: The references mentioned by the reviewer have been incorporated into the manuscript (L282, 285).**

RC2: L.291-332, please provide the number of samples per LME you used for these analyses.
**Response: The number of data samples collected for each Large Marine Ecosystem (LME) region has been added. See Figure S1 and Fig. 3 caption (L315-318).**

RC2: Did you also consider the diversity of methods used per region? I can imagine it also increases if areas are studied using multiple techniques, i.e., different size classes, etc.
**Response: Indeed, the provenance (i.e., sourceArchive column) and the nature of the data (i.e., basisOfRecords) can also affect the observed diversity. We have updated the text to address this concern (L293-301). See also Figure S3.**

RC2: L.378-380 see also Wassmann et al. 2015 (https://doi.org/10.1016/j.pocean.2015.06.011) on Panarctic species advection
**Response: We have modified the text to take account the comment of the referee (L393-394).**

RC2: L.388/389 and, significantly, that it could potentially stimulate cyst germination (see Anderson et al. 2021 https://doi.org/10.1073/pnas.2107387118)
**Response: We have modified the text to take account the comment of the referee (L410-411).**

RC2: L.469-502 Appreciate this analysis that contextualizes sampling bias vs. ecological signals.
**Response: We thank the referee for his comment.**

---

## Author Response (AR2)

**Dear Editor,**

**Thank you very much for your valuable suggestions and comments on our manuscript. We have carefully reviewed the feedback and have made the necessary corrections in line with your suggestions. The new revisions are highlighted in green, while the previous modifications remain highlighted in yellow for clarity.**

**We appreciate your time and consideration, and we hope that the changes meet your expectations. Please feel free to let us know if further revisions are required.**

**Sincerely,**
**Nicolas Schiffrine on behalf of all co-authors**
* * *
**Public justification (visible to the public if the article is accepted and published)**:
Dear authors,

Many thanks for having provided a revised version of your manuscript. I believe all comments from the two reviewers have been addressed and that your manuscript can be accepted for publication.
I just have few requests for minor changes that I would like you to consider.
1) L41: Please do not cite this reference, only a small paragraph on ocean acidification (the fact that the arctic is acidifying more than the rest of the ocean is known for a long time) and prefer citing the few (x3) references cited by Ardyna and Arrigo.
**Response: We have revised the sentence accordingly and now refer to the Arctic Monitoring and Assessment Programme Assessment 2018: *Arctic Ocean Acidification*, which provides a more suitable reference for the rates of acidification in the Arctic (L41).**

2) L211: Please rephrase this new sentence avoiding repetitions: "corresponding to the season with minimal sea ice cover and a better accessibility by ships to the Arctic" or something like this.
**Response: We have reworded this sentence to avoid repetition (L211-212).**

3) L218: Avoid using acronyms that are not used further in the manuscript (SCM). There are potentially more in the text, to check.
**Response: We have removed the acronym "SCM" and checked the manuscript for any other unused acronyms**

4) L258: Isn't it Prochlorococcus marinus?, and L263: elongatus? Perhaps you want to cite only the genera
**Response: The sentences have been rephrased for clarity, and we now explicitly cite only the genus *Prochlorococcus* (L256-259).**

5) L489, and your answer to RC1: you should discuss more in the text the rationale for using this method (also detail LOESS: LOcally Estimated Scatterplot Smoothing)
**Response: We have expanded the discussion to explain the rationale for using the LOESS method, emphasizing its usefulness for capturing non-linear patterns in datasets spanning multiple decades.**